# Depressive Symptomatology and Parenting Stress: Influence on the Social-Emotional Development of Pre-Schoolers in Chile

**DOI:** 10.3390/children8050387

**Published:** 2021-05-13

**Authors:** María Pía Santelices, Francisca Tagle, Nina Immel

**Affiliations:** Escuela de Psicología, Pontificia Universidad Católica de Chile, Santiago 7820436, Chile; fmtagle@uc.cl (F.T.); nfimmel@uc.cl (N.I.)

**Keywords:** pre-schoolers, maternal mental health, social-emotional development, parenting stress

## Abstract

(1) Background: The preschool stage is a period of great psychological changes that requires the support of parents and significant adults for optimal development. Studies show that maternal mental health can be a risk factor in parenting, affecting the social-emotional development of children. (2) Methods: The present study seeks to shed light on the relation between depressive symptoms, parental stress in mothers and social-emotional development of their preschool children, using a total of 123 mother-child dyads with low Social-economic Status (SES). In mothers, depressive symptomatology and level of parental stress were evaluated, as well as social-emotional development in children. A possible mediation effect between maternal depressive symptoms and parenting stress is expected. (3) Results: The results indicate that higher levels of depressive symptoms and parenting stress in mothers relate to greater difficulties in social-emotional development of their preschool children. (4) Conclusions: These results are clinically relevant from the perspective of family therapy: Parents need support to decrease their levels of parenting stress in order not to jeopardise their children’s social-emotional development.

## 1. Introduction

In Chile, 25.7% of women present depressive symptomatology [1]. The term depressive symptomatology (DS) refers to behavioural manifestations that configure a mood disorder characterized by a depressed state [2]. This state can be displayed through a loss of pleasure or interest, changes in appetite, weight, or sleep patterns, a lack of energy, guilt, difficulty in thinking, concentrating, or making decisions, and thoughts of death, among other elements [2]. For depression to be diagnosed, the occurrence of at least five of these symptoms over the last two weeks and a reduction in functioning must be reported: Depressed mood for most of the day, Markedly diminished interest or pleasure in all or almost all activities for most of the day, Significant (>5%) weight gain or loss or decreased or increased appetite, Insomnia (often sleep-maintenance insomnia) or hypersomnia, Psychomotor agitation or retardation observed by others, Fatigue or loss of energy, Feelings of worthlessness or excessive or inappropriate guilt, Diminished ability to think or concentrate or indecisiveness, Recurrent thoughts of death or suicide, a suicide attempt, or a specific plan for committing suicide [2].

Wolf et al. [3] report that depression is three times more prevalent in women during their childrearing period. In line with these results, an international research estimated that 17% of women with young children display depressive symptomatology, which tends to persist throughout the preschool period of their children [4]. Another study reports that depressive symptomatology has a negative impact on nearly all aspects of parenting such as discipline, social modeling, vigilance, and parent-child interaction, including attachment [5].

### 1.1. Maternal Depression and Its Impact on Children’s Social–Emotional Development

Attachment theory suggests that the mother–child relationship is affected by maternal depression because depression impairs maternal sensitivity and availability [6]. When the child experiences emotional stress, it cannot rely on its mother’s care, because the depressive symptomatology generally causes the mother to feel overwhelmed, which reduces her support and co-regulation capacity [7]. Thus, maternal depression can be regarded as a risk factor for social–emotional adjustment and social acceptance [8]. It can even lead to poor social and emotional development along with behavioural problems such as aggressiveness and a contrarian and defiant attitude [9]. Moreover, parental mental health can negatively affect academic attainment of their children during high school and in their future mental health [10]. Fritsch et al. [7] report that 51.9% of children whose mothers suffer from depression display psychiatric symptomatology. Of these children, most display anxious symptoms (62.7%) and/or depressive symptoms (25.9%). In addition, 49.8% display behavioural and emotional problems. Even though these findings were based on clinical samples of school-aged children, they emphasize the need for prevention initiatives aimed at children’s mental health. However, the influence of maternal depressive symptomatology on children’s social–emotional development is not unidirectional. Studies show that certain characteristics of children can influence mothers’ depressive symptomatology: Health problems or irritability in children can affect mothers because their concern for their difficult or infirm children impacts their mood and worsens both maternal symptomatology and the children’s difficulties [11].

### 1.2. Child Social–Emotional Development

Social–Emotional Development refers to both social and emotional skills that manifest themselves in interpersonal relationships and emotional expression [12]. According to Henao and García [13], emotional development includes three relevant aspects that make it possible to identify the emotional competences of pre-schoolers: emotional comprehension, regulation ability and empathy. Social skills allow children to explore rules and gradually understand emotions, manifestations of pro-sociality, and their own interactions with their family and peers [14]. Therefore, emotional and social developments are closely linked. Squires [12] holds that both skills tend to overlap and are activated in relevant situations of the child’s life. Farkas et al. [15] emphasize that young children with an adequate social and emotional development express their emotions appropriately, communicate well with their peers [16,17], and are perceived more positively by their parents [18,19]. In contrast, children with certain difficulties in these areas tend to display disruptive behaviours from an early age onwards, throw tantrums more frequently, and experience negative parent–child communication issues. Given that family is the first context in which children socialize and attain emotional competences, parents have a huge influence on their children’s social–emotional development. Although there is little evidence of social–emotional development differences between boys and girls, an international study showed that boys tend to display more developmental delays in social–emotional development and girls are more emotionally reactive [20].

### 1.3. Parenting Stress in Mothers and Its Repercussions on Children’s Social–Emotional Development

Parenting stress (PS) is understood as a psychological dimension where parents feel overwhelmed by the demands of their parental role [21]. Abidin [6] proposes that parenting stress can be measured in three dimensions: (1) Parental Distress, which refers to the stress perceived by the parent in connection with her personal traits given the demands posed by parenthood; (2) Dysfunctional Mother–Child interaction, which refers to the stress perceived by the mother in the interaction with her child; and (3) Difficult Child, which refers to how easy or difficult it is for the mother to control her child given its behavioural traits [22]. Jackson et al. [23] show that mothers with high levels of parenting-related stress have children who tend to externalize and display behavioural problems. It has been reported that boys are more likely to display such problems [24], which suggests that mothers with male children may tend to experience more parenting stress.

Regarding the impact of parenting stress on children’s social–emotional development, it has been proposed that maternal stress affects overall development [25]. This is consistent with the Early Childhood Longitudinal survey [Encuesta Longitudinal en Primera Infancia en Chile, ELPI] data [26], which suggests a link between parenting stress indexes and children’s learning (cognitive development), with risks being greater in the case of clinical stress levels. The findings of Waters [27] also indicate a strong link between parenting stress and children’s stress.

### 1.4. Relation between Maternal Depressive Symptomatology and Stress

The stress model proposed by Crnic & Low [28] explains the link between stress and health, revealing important connections between stress, disease (in this case depression), and psychological well-being. Life events and stressors as a whole, along with personal, social, and material resources, are regarded as key factors that lead a person to assess experiences as either challenges or threats. The authors emphasize the importance of coping, as individuals with limited coping strategies are more likely to become stressed [28].

In another study, Wang et al. [29] examined the relationship between stress and depression of university students and showed a significant positive correlation, indicating that stress is an important factor in the generation of depression symptoms.

International evidence supports the association between parenting stress and maternal depressive symptomatology [30,31]. Even though these mothers sometimes are sensitive (properly recognizing and interpreting signals), on other occasions they display an intrusive or rejecting behaviour, thus unpredictable. This may lead the child to internalize mistrust towards the mother, showing defensive behaviour, and a negative emotional core, characterized by sadness and anger [32,33].

### 1.5. Children’s Social–Emotional Development and Maternal Mental Health (Depressive Symptomatology and Parental Stress)

As mentioned earlier, the family in which the child grows up exerts the strongest influence on his/her development. A study conducted in Chile stresses the importance of social–emotional development competences and positive learning in children who have fewer opportunities and are at social risk, due to belonging to vulnerable groups [34]. In addition, according to ELPI [26] data, the first income quintile displays the highest percentage of clinical parenting stress in mothers (28.6%), followed by quintile 2 (24.45%). The reason for people living in lower-SES contexts displaying more depressive symptomatology is not only due to poverty, but also to contextual stressors and the quality of their environment. Issues such as low schooling rates, single-parent families, precarious jobs, informal or insecure housing, and limited family support, lead to a higher vulnerability [35]. In such contexts, many mothers may tend to feel hopeless and regard motherhood negatively due to its unexpected nature, which they accept with resignation and anguish, in some cases leading to depression. Nair et al. [36] examined how psychosocial risk factors accumulated over the first 18 months of a child’s life influenced changes in parenting attitudes and children’s development. They found that, as the number of risks increased, women reported more stress related to caring for their children. However, they did not find effects of psychosocial risks on children’s motor and mental development. This might be due to the young age of the children in this study. As the children age into preschool and school years, the variability in psychosocial risk may become more apparent in their development. Huang et al. [37] studied the impact of parenting stress, social support, and depression in a vulnerable population (ethnic minority adolescent mothers) on child development and found that higher levels of parenting stress and less perceived social support were associated with higher levels of maternal depression which in turn was associated with more developmental delays in their infant children. Their study indicated that depression mediated the relation between parenting stress and later child outcomes.

The present study aims to examine the influence of mothers’ mental health (nonclinical sample) on their children’s social-emotional development. The first specific objective is to examine the associations among maternal depressive symptomatology and parenting stress in relation to difficulties in pre-schoolers’ socioemotional development. The second specific objective is to examine these associations controlling for gender and socioeconomic status (SES). The third specific objective is to examine whether parenting stress mediates the relation between maternal depressive symptomatology and child social–emotional development. Analyses will be carried out using a cross-sectional sample. For this reason, the results, particularly from the mediation analysis, will be interpreted in terms of association. The present study could contribute to the validation of a dyadic/family focus in health care models and inform local efforts to design preventive interventions aimed at benefiting pre-schoolers’ current and future mental health.

## 2. Materials and Methods

### 2.1. Participants

The sample was obtained from seven public preschools in Santiago de Chile, which are mainly attended by families with psychosocial vulnerability. A convenience sampling method was used: the researchers selected an environment (preschools) to study individuals from the population who were available by chance and who voluntarily agreed to participate. To be included in the sample, children had to attend preschool regularly, had no diagnosed pathologies (special educational needs), gave their assent to participate, and whose parents agreed to take part in the study by signing an informed consent document. This led to a sample of 123 mothers with an average age of 29.77 years (Standard Deviation (SD) = 6.39, ranging from 19 to 47 years) and their children. With respect to the children, 48.6% were girls (54). Their average age was 44.68 months (SD = 3.62, ranging from 36 to 53 months). The majority of participants belong to mid SES environments. Therefore, the sample is more representative of mid and lower socioeconomic status, since one of the requirements to access free public preschools in Chile is to have low income, and all of the participants in the study attend these (Table 1).

### 2.2. Instruments

Beck Depression Inventory (BDI; [38]). The BDI measures the behavioural manifestations of depression and was created to detect and quantify depressive symptoms. It is a self-report instrument composed of 21 categories/items reflecting symptoms or attitudes derived from clinical observation that are later scored from 0 to 3, depending on severity (neutral to maximum). Scores range from 0 to 63 and reflect the perceived gravity of the respondent’s depression. In the present study the following cut-off scores are used: 0–9 (no depression or minimum depression), 10–18 (mild depression), 19–29 (moderate depression), and 30–63 (severe depression; [39]). The BDI has shown strong internal consistency in both clinical and nonclinical samples, with a Cronbach’s alpha of approximately 0.92 [40]. In this current study, BDI Cronbach’s alpha was 0.87.

Parental Stress Index-Short Form (PSI-SF; [6]). The PSI-SF provides a quick measure of stress levels, based on the assumption that stress is due to situational variables, parental characteristics, and/or behavioural traits of children related to the parenting role. It is composed of 36 items with a Likert-type response scale. Scores are calculated for each of the three subscales (12 items each): Parental Distress (PD), Dysfunctional Parent–Child Interaction (DI), and Difficult Child (DC). The Parental Distress (PD) subscale explores the distress experienced by caregivers when performing a parenting role. The Dysfunctional Parents–Child Interaction (DI) subscale explores the parent’s perception about their child, expectations, and the reinforcement the child gives back to their parents in their parenting role. The Difficult Child (DC) subscale centres on how easy or difficult the parent perceives the child based on his/her behaviour [22]. Over 30 on the subscales reflects high levels of stress [6]. The sum of these three subscales yields a score that indicates the respondent’s total stress level [22]. The PSI has undergone different validity and reliability tests; especially, the short version in Spanish has demonstrated excellent reliability (Cronbach’s alpha over 0.95). In this study all subscales showed good reliability: α = 0.831 for Parenting Stress, α = 0.819 for Difficult Interaction, and α = 0.835 for Difficult Child. The cut-off for the original scale is the 85th percentile [6]; however, a study using the Spanish version suggested lower percentile scores (73rd–77th). For the current study, PSI Cronbach’s alpha was of 0.90.

Ages & Stages Questionnaires: Social-Emotional (ASQ-SE; [40]). The ASQ-SE’s objective is to identify children at risk of manifesting social or emotional problems, without diagnosing disorders. It consists of 30 numerically scored items, completed by the parents or the main caregiver. It was constructed for children aged 1–72 months, with different versions for different ages. The areas covered by the questionnaire include self-regulation, communication, adherence, adaptive functioning, autonomy, affectivity, and social interaction. Higher scores reflect an increased likelihood of displaying social-emotional difficulties. The cut-off score varies depending on the infant’s age (in months), with scores above the cut-off reflecting less social–emotional development. The instrument has shown good reliability (Cronbach’s alpha = 0.94).

Socio-demographic Questionnaire (for mothers). Developed by the research team, the socio-demographic questionnaire provides information to characterize the sample, covering both familial and individual aspects of the pre-schoolers and parents that take part in the study. Socioeconomic variables were assessed by Adimark guidelines, using variables such as educational level and occupational level. For analysis purposes, the variable ESOMAR will be considered as a socioeconomic status (SES) continuous variable.

### 2.3. Procedure and Data Analysis

After contacting several preschools and proposing the project, parents were invited to take part in the study voluntarily. Adult participants were asked to sign an informed consent document, while children had to give their verbal assent before being evaluated.

Data were coded and statistically analysed using R. Missing data were handled by listwise deletion and only cases with complete data were included. The level of significance used in this study to analyse the data obtained was 0.05. As studies have shown that girls are more emotionally reactive in social–emotional development and boys tend to display more developmental delays [16], child gender was used as a statistical control variable in this study. In the first step, data was analysed descriptively. Next, a correlation analysis was implemented to explore if there were any significant associations between depressive symptomatology and parenting stress, and social–emotional problems. In order to deepen the associations between variables, parenting stress subscale’s scores were used in the correlation matrix (PSI_PD: parental distress, PSI_DI: difficult interaction, PSI_DC: difficult child). Afterwards, linear regression models were generated to explore the magnitude of the association between parenting stress and depressive symptomatology and social–emotional development in children, after controlling for gender and socioeconomic status. Finally, to evaluate if the relationship between depressive symptomatology and socioemotional development is mediated by parenting stress, a concurrent mediation analysis was conducted. The indirect effect was estimated using a bootstrapping approach, which shows a better estimation of the indirect effect of mediation models [41]. Because the sample is cross-sectional, and mediation processes occur longitudinally, the effect sizes are expected to be smaller. For this reason, the paths’ coefficients and significance are likely to be smaller, or not significant at all, which could be a problem when testing the previous steps that support a mediation analysis [42]. However, new approaches indicate that the non-significance in step 1 [42] would not be a problem when there are reasons to believe that the effect size is small [43].

## 3. Results

### 3.1. Descriptive Analysis

Means and standard deviations for maternal depression and parenting stress as well as child social–emotional development are shown in Table 2. Mothers displayed an average depressive symptomatology index of 7.77, reflecting few, if any, symptoms of depression.

Concerning parenting stress, results showed an intermediate stress level on the total scale. Regarding the instrument’s subscales, the participants report an average score of 27 for parental distress (PSI_PD), indicating a heightened stress level. Parenting stress perceived by the mother due to a dysfunctional interaction between her and her child (PSI_DI), was at an intermediate stress level. Mothers’ stress due to finding it difficult to control their children, given the child’s characteristics (PSI_DC), reached an average of 26, indicating mid–high stress levels.

In our sample, the mean score of the ASQ-SE was 54, reflecting adequate socio–emotional development.

### 3.2. Correlation Analysis

Results of the correlation analysis can be found in Table 3. Based on these correlations, we can state that depressive symptomatology in mothers was not associated with higher social–emotional development problems in children when there is no variable control. On the other hand, depressive symptomatology is positively associated with PSI total, specifically with parenting stress and the stress related to having a difficult child, given the child’s characteristics (DC).

### 3.3. Association between Parenting Stress and Depressive Symptomatology with Social-Emotional Development Risk in Pre-Schoolers

In order to determine the magnitude of the relation between maternal depressive symptomatology and parenting stress and child social–emotional development, five linear regression models are used. Unstandardized regression coefficients (*b* values) and general model fit (R^2^) are shown in Table 4.

Model 1 only included maternal depressive symptomatology (BDI-II) as a predictive variable and child gender and socioeconomic status as control variable. Models 2 to 4 each added one of the parenting stress subscales (PSI_PD; PSI_DI; PSI_DC) to the analysis and Model 5 included all three subscales. Results of models from 2 to 4 show that each of the parenting stress components relates to social emotional problems, approximately 1.7 points higher in the social emotional development scale in each dimension of the PSI scale. When each dimension of the PSI scale is estimated controlling by the other two dimensions (model 5), it is observed that parental stress is no longer significant (*p* = 0.17). This means that the association between parental stress and social emotional development may be explained mainly by having difficult children or a dysfunctional relationship with the child. Our results show that depression (BDI) was not associated with social emotional problems after controlling by gender, SES, parental stress, dysfunctional interaction and difficult child.

### 3.4. Mediation Analysis

A mediation analysis was made with the object of finding if the effect of depressive symptomatology is mediated by parenting stress following indications made by Baron & Kenny [42]. The PSI total scale was used in this mediation in order to focus on the total parenting stress level and measure its association with depressive symptomatology. First, the existence of a relationship between depressive symptomatology and social–emotional development was tested (*b* = 0.72, SE = 0.40, *p* = 0.072). Next, depressive symptomatology was shown to be associated with parental stress (*b* = 1.04, SE = 0.23, *p* < 0.000). After that, it was shown that parenting stress explained social–emotional development more than maternal depression (BDI) (*b* = 0.78, SE = 0.15, *p* < 0.000), and that the path between maternal depressive symptomatology and social–emotional development is significant (*b* = −0.09, SE = 0.37, *p* < 0.818). Given that this relationship is no longer significant, it is suggested that the effect of depressive symptoms on ASQ-SE is fully mediated by parenting stress (Table 5). This finding should be interpretated with caution due to the non-significance of the first step, and the cross-sectional nature of the data. Although the first step is partially significant, this does not always mean the absence of mediation [43]. Mediational analyses are mainly longitudinal due to the exploring process. Hence, the limitations with concurrent mediations due to the cross-sectional data mean that possibly there is no time to produce the mediation process completely. For this matter, the findings will be handled as mediation possible effects, and must be confirmed in later studies. The indirect effect was estimated using the bootstrapping approach, which is best at estimating this effect [41]. The indirect effect of stress-mediated depressive symptoms was significant (*b* = 0.8 *p* = 0.001). That is, people who score 10 points or more in the depressive scale compared toothers, obtain a mean of 8 points more in social–emotional problems, due to the possible effect caused by depressive symptomatology on social–emotional development through parenting stress.

## 4. Discussion

The assessment of maternal mental health in nonclinical contexts showed that most mothers display subclinical ranges of depressive symptomatology and intermediate stress levels. Regarding socio-emotional development, the participating children in the study showed adequate socio–emotional development. These results compared to local statistics obtained in the ELPI survey [26] may constitute a type of research bias, considering that our results contrast with the results obtained in the general population study. This could be because the latter studies children who attend preschools at an early age, which also implies a positive level of resource for mothers.

In this study, there was no direct association found between depressive symptomatology and socio–emotional development in children, but a direct association between parenting stress and social–emotional development risk was found. First, it can be observed that the symptoms of both depression and parenting stress are closely connected, especially when managing their child´s behaviour due to his/her own characteristics (DC), which has been shown in other studies where maternal depression is associated to the perception of the child as a difficult child [9]. As other studies have shown, negative maternal perceptions are related to a greater likelihood of physical punishment, which in turn leads to more externalization of problems by children [9,28].

When analyzing the subscales of the Parental Stress Index, only PSI_DI (difficult interaction) and PSI_DC (difficult child) were significantly associated with child social–emotional development, which could indicate that the aspects that most affect child development are the mother’s perception that her child is difficult to manage and the perception of having a conflictive interaction with him/her. A positive relation between the Parental Stress Index subscales and social emotional development is also shown. However, when analyzing the three dimensions at the same time, the relation with parental stress (PD) is explained by the other two subscales (DC and DI). This result should be interpreted with caution due to multicollinearity concerns related to including three highly correlated subscales from the same measure.

Depressive symptomatology may cause mothers to have low self-esteem, display decreased vital energy and lose interest in pleasurable activities, coexisting with parenting stress, all of which could make them feel overwhelmed and even unproductive in their parenting role [6]. Our results suggest that the relation between maternal depressive symptomology and child social–emotional development may be mediated by parenting stress. While the first step in the mediation process proposed by Baron and Kenny [44] was not met, as the association between maternal depressive symptomology and child social–emotional development only approached statistical significance, more recent recommendations provided by Shrout and Bolger [45] suggest that our analyses indicate mediation. The lack of significance in the first step may be due to the cross-sectional nature of the data (low effect size of the process explored in the mediation). Given this limitation, these mediation results are interpreted as exploratory and should be examined in future longitudinal studies with similar samples. Our results may suggest that parenting stress is more reflective of parental behaviour and has a more direct effect on child socio–emotional development than depressive symptomatology. Recent studies also support this idea as parents reporting greater levels of parenting stress have children with higher levels of socioemotional development problems than parents reporting lower levels of parenting stress [46]. Starting from this, it is important to highlight that the familial context of the dyad and the stress that this context produces has a stronger influence on child development outcomes than maternal depression [47].

## 5. Conclusions

Given the importance of the mother–child bond for the social and emotional development of the child, it is crucial to highlight the necessity of helping and supporting mothers suffering from both depressive symptomatology and parenting stress, in order to decrease their children’s risk of delays in their social–emotional development.

### 5.1. Implications for Family Therapy Practice

In our study we show that parenting stress may be associated with the effect of maternal depressive symptomatology on child social–emotional development. A mother with depression is more likely to display maternal stress, and her stress affects her child’s social–emotional development. Therefore, the treatment of the maternal depressive symptomatology not only benefits the mother in relation to the remission of symptoms, but also affects the level of stress she experiences in the upbringing of her child. As our results show, children could be directly affected by maternal stress, therefore it is important to include both, mothers and children, in the therapeutic process. By focussing not only on the depressive symptomatology but also on the stress that the mother could be feeling and the link between mother and child, family therapy helps to improve the relationship.

However, parents should be able to seek professional support not only when experiencing depressive symptomatology, but when feeling overwhelmed by the context, having a dysfunctional relationship with their child, or when having a difficult child. Having family problems, work problems, or difficulties to manage and raising a child, among other causes that could affect the amount of stress experienced, are enough to cause problems within the relationship between the parent–child. Therapy could help the mother find a space where she can express herself and learn how to manage stress in a way which doesn´t affect her child. In mother–child sessions the dyad can also learn to spend quality time together through bonding therapy.

The mother’s perception of her child and of the interaction with her child is another important factor. When reviewing the PSI subscales, regarding the dysfunctional interaction subscale, a negative perception of the interaction is by itself a risk factor to the children’s social-emotional development, because negative emotions and a poor perception of the child are conveyed, thus reducing the child’s self-esteem [48]. At the same time, the lower a child’s social–emotional development, the poorer the mother’s perception of her child [16,17]. This indicates that children’s emotional and social difficulties are not only linked to the symptoms affecting their mothers’ mood but also to their mothers’ perception of parenting and how they exercise it. Our findings suggest that if the mother’s perceptions were altered, this might lead to more positive behaviour, directly affecting their children’s social–emotional development.

The present study provides indicators for the importance of the early detection of depressive symptomatology in women during their childrearing period and the necessity of preventive family interventions to reduce parenting stress and improve mother’s perception of their children. Given the consequences of a poor social–emotional development, both for the child and for the interaction with his/her social context and his/her future possibilities, interventions to help prevent parenting stress, and thereby improving present and future mental health of pre-schoolers, is crucial.

Above all, this research validates the importance of conceptualizing that certain problems are generated in a family context. When a mother has depressive symptoms and in consequence higher parenting stress, she changes her modes of interaction. To prevent consequences in the development of her preschool children, these interaction modes must be reorganized. Family therapy, either ambulatory or as part of other health care services, is especially apt to provide help for such situations.

### 5.2. Limitations

Even though it is a cross-sectional study that provides descriptive and analytic information, the current study does not reveal how and to what extent an intervention for mothers would improves the social–emotional development of their preschool-aged children. Another limitation is that only self-report instruments were employed, which may have influenced our results because mothers’ responses to the ASQ-SE were biased by their perceptions. This led to a loss of information on the children’s social-emotional development because there were no observations of the children or of the dyadic interaction, which would have increased objectivity. It must also be considered that mean depressive symptomatology was below clinical levels, and parenting stress was lower than expected, thereby limiting statistical results. Another limitation is that the data only pointed at possible mediation. The results in this study indicate a mediating role of parenting stress in the association between maternal depressive symptomatology and child social–emotional development; however, this mediation must be further explored.

In future studies, being able to identify connections between maternal mental health and observable child and interaction variables would result in relevant findings for designing preventive interventions aimed at the early childhood stage. It would also be interesting to study the interaction between parenting stress and depressive symptomatology and carry out more in-depth research with a gender approach.

## Figures and Tables

**Table 1 children-08-00387-t001:** Participants.

		Mean or *n*	SD	Min	Max	SE
Children	Children’s age in months	44.68	3.62	36	53	0.34
	Gender female	54(49%)				
Mothers	Adult´s age in years	29.77	6.39	19	47	0.6
Familial SES	D	13(12%)				
	CB	39(35%)				
	CA	36(32%)				
	B	17(15%)				
	A	7(6%)				

Standard Deviation (SD), Standard Error (SE), Familial SES: D (mid-low), CB (mid), CA (mid-high), B (high), A (very high).

**Table 2 children-08-00387-t002:** Means (Standard Deviations) for maternal and child variables.

	Mean	SD	SE	Min	Max
Depression (BDI)	7.77	6.52	0.61	0	40
Total PSI	71.69	19.45	1.83	40	155
Social-Emotional Development (ASQ-SE)	53.76	27.72	2.61	0	155
Parental Distress (PD)	27.33	9.02	0.85	12	55
Dysfunctional Interaction (DI)	18.69	6.84	0.64	12	51
Difficult Child (DC)	25.67	7.81	0.73	13	49

PSI: Parental Stress Index.

**Table 3 children-08-00387-t003:** Correlation between depression, parental stress and child social-emotional development.

	Parental Distress (PD)	Dysfunctional Interaction (DI)	Difficult Child (DC)	Total PSI	Depression (BDI)	Social-Emotional Development (ASQ-SE)	SES
Parental Distress (PD)	-	0.488 ***	0.558 ***	0.860 ***	0.461 ***	0.403 ***	−0.236 *
Dysfunctional Interaction (DI)		-	0.469 ***	0.767 ***	0.149	0.420 ***	−0.251 **
Difficult Child (DC)			-	0.826 ***	0.207 *	0.502 ***	−0.053
Total PSI				-	0.349 ***	0.536 ***	−0.219 *
Depression (BDI)					-	0.17	−0.151
Social-Emotional Development (ASQ-SE)						-	−0.05
SES							-

* *p* < 0,05; ** *p* < 0,01; *** *p* < 0,001; SES: Social-Economical Status.

**Table 4 children-08-00387-t004:** *b*-Values and R-Squared of the five Linear Regression Models.

	Model 1			Model 2			Model 3			Model 4			Model 5		
	*b*	SE	*p*	*b*	SE	*p*	*b*	SE	*p*	*b*	SE	*p*	*b*	SE	*p*
Constant	48.04	6.47	<0.001	27.54	7.38	<0.001	32.34	6.68	<0.001	26.11	6.71	<0.001	18.91	7.01	0.008
Depression (BDI)	0.69	0.43	0.112	−0.23	0.44	0.598	0.44	0.4	0.269	0.3	0.38	0.431	0	0.41	0.993
Female	3.57	5.51	0.519	7.29	5.11	0.156	5.96	5.02	0.238	3.28	4.79	0.495	5.86	4.72	0.218
SES	−0.79	2.54	0.756	1.18	2.36	0.617	1.81	2.36	0.445	−0.44	2.21	0.841	1.42	2.22	0.523
Parental Distress (PD)				1.46	0.32	<0.001							0.49	0.36	0.167
Dysfunctional Interaction (DI)							1.79	0.37	<0.001				0.91	0.39	0.023
Difficult Child (DC)										1.8	0.3	<0.001	1.17	0.36	0.002
R2/R2 adjusted	0.037/0.010			0.201/0.171			0.216/0.186			0.281/0.254			0.341/0.303		

**Table 5 children-08-00387-t005:** Mediation analysis.

Dependent Variable	Independent Variable	*b*	SE	*p*
PSI total	Depression (BDI)	1.042	0.232	0
Social-Emotional Development (ASQ-SE)	PSI total	0.775	0.146	0
Social-Emotional Development (ASQ-SE)	Depression (BDI)	−0.085	0.371	0.818
Indirect effect		0.807	0.251	0.001
Total effect		0.722	0.408	0.077

## Data Availability

Data sharing is not applicable to this article due to privacy issues.

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
