# Peer review of "Depressive Symptomatology and Parenting Stress: Influence on the Social-Emotional Development of Pre-Schoolers in Chile"

_children, 2021, doi:10.3390/children8050387_

Round 1

Reviewer 1 Report

INTRODUCTION

- line 118: replace “in” with “is”

-line 120 and line 124: please update references

- line 153-154: this sentence is a repetition of previous one

- line 155 to 158: please review English writing

- line 155: I would not describe correlation analysis as a specific aim. I would suggest authors to present correlation as preliminary analysis and focus their study’s aims on regression and mediation models.

- line 156: if study design is cross-sectional instead of longitudinal I suggest authors to avoid terms such as “predictive value”, present analysis in terms of associations.

- line 159: this is a very relevant point: authors talk about moderation and mediation as the same thing, while they’re not! Even if not discussion theoretical reasons to perform mediation or moderation analysis, authors should carefully review their study, being consisted in terminology and having understood the methods adopted.

- line 236-237: please, clarify whether you conducted moderation or mediation analysis and describe the statistical method adopted (which test and statistical procedure?)

- line 260 and after in manuscript: authors should clarify whether the design of the study is longitudinal. If not, I strongly suggest to replace terms suggesting temporal relationship, such as "predictors", with proper correlational language (i.e., association, associated variables)

- line 284: I suggest to include a figure presenting mediation analysis.

Author Response

Reviewer 1

INTRODUCTION

- line 118: replace “in” with “is”

Addressed:

“indicating that stress is an important factor in the generation of depression symptoms.”

-line 120 and line 124: please update references

Addressed:

Reid & Taylor, 2015

Lieberman & Van Horn, 2011

- line 153-154: this sentence is a repetition of previous one

Addressed. The sentence was erased.

“The present study aims to identify the possible effect of mothers' mental health (nonclinical sample) on their children's social-emotional development, exploring how maternal depressive symptomatology and parenting stress are related to social-emotional development of preschool children.”

- line 155 to 158: please review English writing

Addressed. Reviewed and corrected.

“(…) parenting stress associates with difficulties in their children´s socioemotional development. The second specific objective is to identify the association between parenting stress and depressive symptomatology in problems related with children´s socioemotional development controlled by gender and socioeconomic status (SES).”

- line 155: I would not describe correlation analysis as a specific aim. I would suggest authors to present correlation as preliminary analysis and focus their study’s aims on regression and mediation models.

Addressed. Changed to associates and not correlates:

“The first specific objective is to know if maternal depressive symptomatology and parenting stress associates with difficulties in their children´s socioemotional development.”

- line 156: if study design is cross-sectional instead of longitudinal I suggest authors to avoid terms such as “predictive value”, present analysis in terms of associations.

Addressed. Erased “predictive value” and replaced for association:

“(…) to identify the association between parenting stress (…)”.

- line 159: this is a very relevant point: authors talk about moderation and mediation as the same thing, while they’re not! Even if not discussion theoretical reasons to perform mediation or moderation analysis, authors should carefully review their study, being consisted in terminology and having understood the methods adopted.

Addressed. Corrected.

“(…) is mediated by parenting stress, a concurrent mediation analysis was made (…)”.

- line 236-237: please, clarify whether you conducted moderation or mediation analysis and describe the statistical method adopted (which test and statistical procedure?)

Addressed. Clarified.

“(…) a concurrent mediation analysis was made. To collect evidence that show the existence of mediation, in first place there were used the steps proposed by the authors Baron & Kenny (1986). The indirect effect was estimated using bootstrapping approach. This approach has a better estimation of the indirect effect of mediation models (Hayes, 2017).”

- line 260 and after in manuscript: authors should clarify whether the design of the study is longitudinal. If not, I strongly suggest to replace terms suggesting temporal relationship, such as "predictors", with proper correlational language (i.e., association, associated variables)

Addressed. Changed and replaced every term that could suggest temporal relation, to a proper correlation language through all the manuscript.

- line 284: I suggest to include a figure presenting mediation analysis.

Table 6 presents the mediation analysis.

Reviewer 2 Report

I was surprised to read the responses form authors. While some of the issues raised by both reviewers have been addressed, many have not. Responses such as "Thank you for the comment. Nevertheless, we could not address this point due to the lack of time given to correct and add this information to the study." are hard to justify. I encourage the authors to revise the work according to the feedback provided. The work as currently stands is not complete. Only one example is provided here given that many points raised by reviewers have not been sufficiently addressed. For example, a mediation analysis should be performed.

Author Response

Reviewer 2

I was surprised to read the responses form authors. While some of the issues raised by both reviewers have been addressed, many have not. Responses such as "Thank you for the comment. Nevertheless, we could not address this point due to the lack of time given to correct and add this information to the study." are hard to justify. I encourage the authors to revise the work according to the feedback provided. The work as currently stands is not complete. Only one example is provided here given that many points raised by reviewers have not been sufficiently addressed. For example, a mediation analysis should be performed.

Addressed.

We performed a mediation analysis (table 6), and also, we addressed the points we could not at first place. It has new analyses, regression models, and new data that is analyzed further in the discussion.

Reviewer 3 Report

The authors aim to explore the relationship between maternal depression, parenting stress, and social-emotional development of pre-schoolers. Whilst parental support is needed for adequate child development, maternal depression can lead to problems in parenting and poor child social-emotional development. The present study evaluated depressive symptoms and parenting stress in mothers and social-emotional development in pre-school children from 123 families with mid and low socio-economic status in Chile. The findings suggest that the relationship between maternal depressive symptoms and child social-emotional development is possibly mediated via parenting stress. These findings shed light on the role maternal support plays in pre-school child social-emotional development and may pave the way to developing new interventions to reduce maternal depressive symptoms and parenting stress and thereby prevent social-emotional problems in pre-schoolers.  The research question investigated in this study is of importance and contributes to existing evidence. The manuscript is written in a coherent manner but requires moderate language editing. Please see my comments below.  

Title

  1. Line 2-4: “parental stress” - should be changed to “parenting stress”. Please change where appropriate throughout the manuscript. I would also remove the word “Their”.

Abstract

  1. Lines 16-17: it should be stated here that the study involved participants with low SES.
  2. Lines 17-19: the results should include a statement about the possible mediation effect.

Keywords

  1. Line 23: Please correctSocio-emotional” to “social-emotional”.

Introduction

  1. Line 70: I would add a sub-heading “maternal depression and its impact on children’s social-emotional development
  2. Lines 31-33: “For depression to be diagnosed… must be reported” - I would outline the depression criteria used by DSM-5.
  3. Line 34: “Wolf… (2002)” - the names of the second authors onwards can be replaced with “and colleagues” or “et al.”. Please replace where appropriate throughout the manuscript.
  4. Lines 60-63: “On the other hand…studied the interdependence between … and their results suggested…” - this sentence should be rephrased to something along the lines: “ On the other hand, the study by Sacchi and colleagues suggested that…”.
  5. Lines 102-108: “…maternal stress stimulates overall development …(cognitive development)” - according to this sentence it sounds as if maternal stress is good for child development which is the opposite to what is argued in this paragraph and the manuscript. I would change the sequence of the sentences so that the arguments that are consistent with the hypothesis of the present study will be first and then add information from the other studies while clarifying the differences.
  6. Lines 156-157: “…predictive value…” - predictive value has a different meaning which is not suitable here, therefore I would not use it in this context. I would replace this sentence with “to identify the extent to which parental stress and depressive symptomatology predict problems...”.
  7. Lines 159-160: this sentence belongs to the preceding paragraph.

Materials and methods

  1. Table 1: “Familiar SES” did you mean “familial SES”?
  2. Lines 195 and 258: “paternal distress” please change to “parental distress”.
  3. Line 202: “(Abadin, 1990)” did you mean Abidin?
  4. Line 237: “…a concurrent moderation analysis was made” please change moderation to mediation.
  5. Line 254: “…is not linked to higher social-emotional development in children.” Did you mean “higher social-emotional development problems in children”?
  6. Line 269: “Controlling by SES…”. Please clarify whether you also controlled for gender.
  7. Line 269-273: “…disappeared when including the parental stress subscales…“ - according to Table 4 it seems as if depressive symptoms do not significantly predict ASQ-SE (model 1 p value 0.112), therefore it is unclear how introducing the additional variables made it insignificant in models 2-5 if it was like this in the first place. Please clarify.
  8. Line 271: “(B=0.370, … respectively)”. I would suggest adding that these values are related to the strength of the associations between the PSI sub-scales and child social-emotional development.
  9. Lines 277-280: “…effects in the bivariate analyses” please note where these data are shown.
  10. Lines 280-282: Please outline the results of this analysis.
  11. Lines 287-288: Please clarify how this analysis is different from linear regression models 6 and 7 given it yielded different results, was this because you did not control for gender and SES?
  12. Line 292: following “development is significant (b = -0.09, se = 0.37, p < .818)” I would add that given this relationship is no longer significant it is suggested that the effect of depressive symptoms on ASQ-SE is fully mediated by parenting stress.

Discussion

  1. Lines 294-295: “…there is no time to produce…” - please clarify.
  2. Line 317: “… the general population study” - please add a reference.
  3. Lines 329-336: paragraph starting in line 336 should be a continuation of the preceding sentences.
  4. Lines 333-336: these sentences are unclear, please clarify and add references to support your arguments.
  5. Lines 358-359: “…parental stress may influence the effect of maternal depressive symptomatology” this is incorrect as you show mediation effect and not moderation effect. Please correct.
  6. Lines 412-414: “this led to a loss of information… objectivity” - please clarify the meaning of these sentences.
  7. Line 416: “Another limitation is the lack of a mediating model.” - I would change this sentence to state that the data only point at possible mediation.

General comment

  1. The manuscript would benefit from moderate language editing including typos and the use of the correct prepositions.

Author Response

Reviewer 3

Title:

  1. Line 2-4: “parental stress” - should be changed to “parenting stress”. Please change where appropriate throughout the manuscript. I would also remove the word “Their”.

Addressed.

“Depressive Symptomatology and Parenting Stress: Their Influence on the Social-Emotional Development of Pre-schoolers in Chile”

Abstract:

  1. Lines 16-17: it should be stated here that the study involved participants with low SES.

Addressed.

“(…) parental stress in mothers and social-emotional development of their preschool children, using a total of 123 mother-child dyads with low SES.”

3.     Lines 17-19: the results should include a statement about the possible mediation effect.

Addressed. It was added:

“It is expected to find a possible mediation effect between maternal depressive symptoms and parenting stress.”

  1. Line 23: Please correct “Socio-emotional” to “social-emotional”.

Addressed. It was changed through all the manuscript.

Introduction:

  1. Line 70: I would add a sub-heading “maternal depression and its impact on children’s social-emotional development

Addressed.

  1. Lines 31-33: “For depression to be diagnosed… must be reported” - I would outline the depression criteria used by DSM-5.

Addressed.

“Depressed mood most of the day, Markedly diminished interest or pleasure in all or almost all activities for most of the day, Significant (> 5%) weight gain or loss or decreased or increased appetite, Insomnia (often sleep-maintenance insomnia) or hypersomnia, Psychomotor agitation or retardation observed by others, Fatigue or loss of energy, Feelings of worthlessness or excessive or inappropriate guilt, Diminished ability to think or concentrate or indecisiveness, Recurrent thoughts of death or suicide, a suicide attempt, or a specific plan for committing suicide (American Psychiatric Association, 2014, p. 104 & 105).”

  1. Line 34: “Wolf… (2002)” - the names of the second authors onwards can be replaced with “and colleagues” or “et al.”. Please replace where appropriate throughout the manuscript.

Addressed.

  1. Lines 60-63: “On the other hand…studied the interdependence between … and their results suggested…” - this sentence should be rephrased to something along the lines: “ On the other hand, the study by Sacchi and colleagues suggested that…”.

Addressed. Changed.

  1. Lines 102-108: “…maternal stress stimulates overall development …(cognitive development)” - according to this sentence it sounds as if maternal stress is good for child development which is the opposite to what is argued in this paragraph and the manuscript. I would change the sequence of the sentences so that the arguments that are consistent with the hypothesis of the present study will be first and then add information from the other studies while clarifying the differences.

Addressed.

“(…) it has been proposed that maternal stress affects overall development (…).”

  1. Lines 156-157: “…predictive value…” - predictive value has a different meaning which is not suitable here, therefore I would not use it in this context. I would replace this sentence with “to identify the extent to which parental stress and depressive symptomatology predict problems...”.

Addressed.

“The second specific objective is to identify the association between parenting stress and depressive symptomatology (…)”

  1. Lines 159-160: this sentence belongs to the preceding paragraph.

Addressed.

Materials and methods:

  1. Table 1: “Familiar SES” did you mean “familial SES”?

Addressed. Corrected.

  1. Lines 195 and 258: “paternal distress” please change to “parental distress”.

Addressed. Changed.

  1. Line 202: “(Abadin, 1990)” did you mean Abidin?

Addressed. Changed.

  1. Line 237: “…a concurrent moderation analysis was made” please change moderation to mediation.

Addressed.

“(…) a concurrent mediation analysis was made (…)”

  1. Line 254: “…is not linked to higher social-emotional development in children.” Did you mean “higher social-emotional development problems in children”?

Addressed.

“(…) mothers is not linked to higher social-emotional development problems in children (…)”

  1. Line 269: “Controlling by SES…”. Please clarify whether you also controlled for gender.

Addressed. This sentence was eliminated due to other analyses that were made.

  1. Line 269-273: “…disappeared when including the parental stress subscales…“ - according to Table 4 it seems as if depressive symptoms do not significantly predict ASQ-SE (model 1 p value 0.112), therefore it is unclear how introducing the additional variables made it insignificant in models 2-5 if it was like this in the first place. Please clarify.

Addressed.

“Results of models from 2 to 4 show that each of the parenting stress components relates to social emotional problems. Nevertheless, with these values we cannot assure that there is one parenting stress component that presents more magnitude than others regarding social emotional problems. In other hand, results show that depression (BDI) would not be associated with social emotional problems. These values are related to the strength of the associations between the PSI sub-scales and child social-emotional development.”

“Regarding Model 5, we can see that when controlling dysfunctional interactions (DI) and child behaviour (DC), the significative relation between parenting stress (PD) and social emotional problems disappears.”

  1. Line 271: “(B=0.370, … respectively)”. I would suggest adding that these values are related to the strength of the associations between the PSI sub-scales and child social-emotional development.

Addressed.

“These values are related to the strength of the associations between the PSI sub-scales and child social-emotional development.”

  1. Lines 277-280: “…effects in the bivariate analyses” please note where these data are shown.

Addressed. This phrase was eliminated because these data no longer were shown in the results.

  1. Lines 280-282: Please outline the results of this analysis.

Addressed. This phrase was eliminated because these data no longer were shown in the results.

  1. Lines 287-288: Please clarify how this analysis is different from linear regression models 6 and 7 given it yielded different results, was this because you did not control for gender and SES?

Addressed.

“In these models, analyses were made with PSI total scores, this means without controlling gender and SES. Through these data it can be shown an association between the PSI total scores and social-emotional development. Also, results show that the ASQ can increase 0.85 whenever the PSI increase 1 point.”

  1. Line 292: following “development is significant (= -0.09, se = 0.37, < .818)” I would add that given this relationship is no longer significant it is suggested that the effect of depressive symptoms on ASQ-SE is fully mediated by parenting stress.

Addressed. Added.

“Given this relationship is no longer significant it is suggested that the effect of depressive symptoms on ASQ-SE is fully mediated by parenting stress.”

Discussion:

  1. Lines 294-295: “…there is no time to produce…” - please clarify.

Addressed.

“Mediational analyses are mainly longitudinal due to the exploring process. Hence, the limitations with concurrent mediations as it is cross-sectional data, possibly there is no time to produce the mediation process completely.”

  1. Line 317: “… the general population study” - please add a reference.

Addressed.

 (ELPI, 2012)

  1. Lines 329-336: paragraph starting in line 336 should be a continuation of the preceding sentences.

Addressed.

  1. Lines 333-336: these sentences are unclear, please clarify and add references to support your arguments.

Addressed.

“Regarding the regression models, these show that the relation between mothers’ depressive symptomatology and socioemotional development may be mediated by parenting stress. This result could significate that parenting stress is more associated with parental behavior, and because of this it has a more direct effect on child development, which does not occur with depressive symptomatology, which is not necessarily associated with a behavioral response.”

  1. Lines 358-359: “…parental stress may influence the effect of maternal depressive symptomatology” this is incorrect as you show mediation effect and not moderation effect. Please correct.

Addressed.

“In our study we could show that parenting stress may be associated with the effect of maternal depressive symptomatology (…)”

  1. Lines 412-414: “this led to a loss of information… objectivity” - please clarify the meaning of these sentences.

Addressed. It means that there were no instruments to increase the objectivity of the dyadic interaction further than what the mother’s own perception.

“there were no observations of the children or of the dyadic interaction, which would have increased objectivity.”

  1. Line 416: “Another limitation is the lack of a mediating model.” - I would change this sentence to state that the data only point at possible mediation.

Addressed.

“Another limitation is that the data only pointed at possible mediation. The results in this study indicate a mediating role of parenting stress in the association between maternal depressive symptomatology and child social-emotional development, however this mediation must be further explored.”

General comment

  1. The manuscript would benefit from moderate language editing including typos and the use of the correct prepositions.

Addressed through all the manuscript.
